# An Aptamer against MNK1 for Non-Small Cell Lung Cancer Treatment

**DOI:** 10.3390/pharmaceutics15041273

**Published:** 2023-04-18

**Authors:** Rebeca Carrión-Marchante, Celia Pinto-Díez, José Ignacio Klett-Mingo, Esther Palacios, Miriam Barragán-Usero, M. Isabel Pérez-Morgado, Manuel Pascual-Mellado, Sonia Alcalá, Laura Ruiz-Cañas, Bruno Sainz, Víctor M. González, M. Elena Martín

**Affiliations:** 1Aptamer Group, Deparment Biochemistry-Research, IRYCIS—Hospital Universitario Ramón y Cajal, 28034 Madrid, Spain; 2Aptus Biotech SL, 28035 Madrid, Spain; 3Department of Cancer, Instituto de Investigaciones-Biomédicas “Alberto Sols” (IIBM), CSIC-UAM, 28034 Madrid, Spain; 4Chronic Diseases and Cancer Area 3—Instituto Ramón y Cajal de Investigación Sanitaria (IRYCIS), 28034 Madrid, Spain; 5Centro de Investigación Biomédica en Red, Área Cáncer—CIBERONC, ISCIII, 28029 Madrid, Spain

**Keywords:** aptamer, MNK1, therapeutic target, non-small cell lung cancer, NSCLC, antitumor

## Abstract

Lung cancer is the leading cause of cancer-related death worldwide. Its late diagnosis and consequently poor survival make necessary the search for new therapeutic targets. The mitogen-activated protein kinase (MAPK)-interacting kinase 1 (MNK1) is overexpressed in lung cancer and correlates with poor overall survival in non-small cell lung cancer (NSCLC) patients. The previously identified and optimized aptamer from our laboratory against MNK1, apMNKQ2, showed promising results as an antitumor drug in breast cancer in vitro and in vivo. Thus, the present study shows the antitumor potential of apMNKQ2 in another type of cancer where MNK1 plays a significant role, such as NSCLC. The effect of apMNKQ2 in lung cancer was studied with viability, toxicity, clonogenic, migration, invasion, and in vivo efficacy assays. Our results show that apMNKQ2 arrests the cell cycle and reduces viability, colony formation, migration, invasion, and epithelial-mesenchymal transition (EMT) processes in NSCLC cells. In addition, apMNKQ2 reduces tumor growth in an A549-cell line NSCLC xenograft model. In summary, targeting MNK1 with a specific aptamer may provide an innovative strategy for lung cancer treatment.

## 1. Introduction

Lung cancer is the leading cause of cancer-related death worldwide and the second most commonly diagnosed tumor among all cancer types [1]. The 5-year survival rate is greater for early-stage lung cancer than for advanced-stage lung cancer. Lung cancer is usually diagnosed in advanced stages, or even when it metastasizes to other areas, which increases therapeutic failure, decreasing patient survival rates [2]. Therefore, the search for new therapies against lung cancer is a healthcare priority.

The mitogen-activated protein kinase (MAPK)-interacting kinases (MNKs) are serine/threonine kinases that are directly activated by an extracellular signal-regulated kinase (ERK) 1/2 or p38 MAP kinases [3,4]. In humans, MNKs comprise a group of four isoforms (MNK1a/b and MNK2a/b) from two genes by alternative splicing. The four isoforms are similar in their N-terminus but differ in their C-terminus. MNK1b was first described in our laboratory [5], and it lacks the MAPK binding site motif present in the C-terminal region of MNK1a. MNK1b has higher basal activity than MNK1a, being independent of ERK1/2 and p38 MAPK activation [6]. MNKs phosphorylate several substrates including eukaryotic initiation factor 4E (eIF4E) [7] and 4G (eIF4G) [8], heterogeneous nuclear ribonucleoprotein A1 (hnRNPA1) [9], cPLA2 [10], Sprouty2 [11], and polypyrimidine-tract binding protein-associated splicing factor (PSF) [12], being eIF4E the best characterized.

The eIF4E improves the translation of proteins involved in proliferation, survival, invasion, and angiogenesis such as cyclin D1, c-Myc, Bcl-2, metalloproteases (MMPs), Snail, vascular endothelial growth factor (VEGF) and fibroblast growth factor (FGF) [13,14,15,16,17]. Phosphorylation of eIF4E on serine 209 by MNKs is associated with tumor progression and poor prognosis, with high levels of phosphorylated eIF4E (p-eIF4E) found in different types of cancer [18]. Similarly, overexpression of MNKs has been found in glioblastoma, lung cancer, hepatocellular carcinoma, ovarian cancer, and breast cancer [19,20,21,22,23,24], and high expression of MNK1 is correlated with poor prognosis in several types of cancers [25]. Of all MNK inhibitors developed thus far [25], some have been evaluated in clinical trials for cancer therapy such as BAY1143269 [26], eFT508 [27], or ETC-206 [28]. Likewise, in MNK1 and MNK2 knockout (KO) mice, while their development is normal, there is a delay in tumor progression in KO mice compared to wild-type mice [29]. Thus, since MNKs are not essential but overexpressed in many tumor types, they represent good therapeutic cancer targets, with theoretically no adverse effects in non-tumor cells.

Aptamers are single-strand nucleic acids (ssDNA or RNA) that bind to different targets by folding into a three-dimensional conformation. They are identified by the SELEX method [30,31], which represents an important tool for the discovery of novel biomarkers of both therapeutic and diagnostic interest. Aptamers offer several advantages over antibodies including their high stability, low immunogenicity, no batch-to-batch variability, small size, short generation time, and quick modification [32]. So far, Macugen^®^ is the only aptamer approved by the FDA for the treatment of age-related macular degeneration [33]. However, several aptamers have entered into clinical trials [34]; some for cancer treatment such as the DNA aptamer AS1411 which targets nucleolin, inducing apoptosis [35], or NOX-A12, an RNA aptamer that inhibits tumor growth by binding to CXCL-12.

We selected an aptamer against MNK1 capable of inhibiting proliferation, migration, and colony formation in MDA-MB-231 breast cancer cells [36]. In order to decrease its size, we obtained four different sequences from the initial aptamer, where apMNKQ2 was the most efficient sequence at inhibiting the proliferation of breast cancer cell lines. Further studies showed that apMNKQ2 inhibits the tumorigenic and metastatic activity of breast cancer cells and reduces tumor growth and metastasis numbers in an in vivo model [37].

Thus, the objective of this work was to evaluate the potential of apMNKQ2 as an antitumor drug. For this purpose, we studied the effect of apMNKQ2 in another type of cancer where MNK1 is overexpressed, specifically lung cancer. Our results show that apMNKQ2 arrests the cell cycle and inhibits the proliferation, colony formation, migration, invasion, and adhesion of lung cancer cell lines, and reduces tumor growth in a mouse xenograft model of lung adenocarcinoma. These results highlight apMNKQ2 as a potential anticancer drug for lung cancer.

## 2. Materials and Methods

### 2.1. Materials

The ssDNA aptamers, apMNKQ2 (TGGGGTGGGCGGGCGGGGGTGGGGGTGGT) and p29 (GCGGTCGACTTAAATGTCCATCTCAAACT) were purchased from Biospring, Frankfurt am Main, Germany. The apMNKQ2 aptamer used in this study does not have any modification. The origin of the rest of the material is indicated in the text.

### 2.2. Cell Culture and Transfection

A549 (human lung adenocarcinoma cell line) and SW900 (human lung squamous cell carcinoma cell line) cells were authenticated in May of 2019, and H460 cells (human lung large cell carcinoma cell line) were authenticated in September of 2022, through the GenePrint^®^ 10 System. SW900 and H460 were cultured in RPMI medium (PAA, Pasching, Austria) containing 10% fetal calf serum (Gibco, Grand Island, New York, NY, USA), 100 U/mL penicillin, 100 μg/mL streptomycin, and 25 μg/mL amphotericin (Sigma, St. Louis, MO, USA). A549 cells were cultured in Dulbecco’s modified Eagle’s medium (DMEM) (Biowest SAS, Nuaillé, France) with 10% fetal calf serum (Gibco, Grand Island, New York, NY, USA), 1% pyruvate, 100 U/mL penicillin, 100 μg/mL streptomycin, and 25 μg/mL amphotericin (Sigma, St. Louis, MO, USA). All cell lines were maintained at 37 °C in a humidified incubator with 5% CO_2_. For cell transfection with aptamers, cells were seeded at different concentrations according to the assay and after 24 h, aptamers were transfected into cells using Lipofectamine^TM^ 2000 (Invitrogen, Carlsbad, CA, USA), according to the manufacturer’s instructions. Cells were incubated at 37 °C in a humidified incubator with 5% CO_2_ until the assay was carried out. Before transfection, aptamers were dissolved in selection buffer (20 mM Tris-HCl pH 7.4, 150 mM NaCl, 1 mM MgCl_2_, and 5 mM KCl), denatured at 90 °C for 10 min and then cooled on ice for 10 min.

### 2.3. Protein and RNA Extraction

For protein extraction, cells were mechanically dissociated and washed once with cold buffer A (20 mM Tris-HCl pH 7.6, 1 mM ethylenediaminetetraacetic acid (EDTA), 1mM dithiothreitol (DTT), 1 mM benzamidine, 1 mM phenylmethylsulfonyl fluoride (PMSF), 10 mM sodium β-glycerophosphate, 10 mM sodium molybdate, 1 mM sodium orthovanadate, 120 mM potassium chloride (KCl), 10 μg/mL antipain, 1 μg/mL pepstatin A, and leupeptin). Then, cells were lysed in buffer A with 1% Tritón X-100 (volume ratio 1:2) and centrifugated for 10 min at 12,000× *g*. Supernatants were used to determine protein concentrations using the BCA kit (ThermoFisher Scientific, Waltham, MA, USA) and then they were stored at −80 °C until use.

For total RNA extraction, cells were trypsinized and centrifugated at 400× *g* for 5 min. Pellets were lysed in NucleoZOL (Macherey-Nagel, Düren, Germany) according to the manufacturer’s instructions. RNA pellets were resuspended in 50 μL of RNase-free water, quantified, and stored at −80 °C until use.

### 2.4. Western Blot

Cell lysates were resolved in 7.5%, 10%, or 12% sodium dodecyl sulphate-polyacrilamide gels (SDS-PAGE) by electrophoresis and transferred onto polyvinylidene difluoride (PVDF) membranes. Membranes were incubated for 1h at room temperature with 5% non-fat milk in PBS and then overnight at 4 °C with monoclonal or polyclonal antibodies. After washing, membranes were incubated with the corresponding peroxidase-conjugated secondary antibodies for 1 h at room temperature and proteins were visualized using Clarity Western ECL Substrate (BioRad, Hercules, CA, USA) and a ChemiDoc^TM^ (BioRad, Hercules, CA, USA). Bands were quantified using ImageLab 6.1 software (BioRad, Hercules, CA, USA). PageRuler Plus Prestained Protein Ladder (ThermoFisher Scientific, Waltham, MA, USA) was the molecular weight marker used in all the experiments. β-actin (Sigma, St. Louis, MO, USA) antibody was used to control the homogeneity of loading. Antibodies used in this work can be found in Appendix A.

### 2.5. Cell Viability (MTT) and Cell Toxicity (LDH) Assays

To study cell viability, 3-(4,5-dimethylthiazol-2-yl)-2,5-diphenyl-2H-tetrazolium bromide (MTT) assays were performed. A549 and H460 cells were plated at 6 × 10^3^ cells/well and SW900 at 10^4^ cells/well in 96-well plates. After 16–24 h, cells were transfected with apMNKQ2 or the unspecific p29 aptamer as described above and incubated at 37 °C in a humidified incubator with 5% CO_2_ for 48 h. Then, the medium was removed to add 100 μL/well of MTT (Sigma, St. Louis, MO, USA) at 1 mg/mL in culture medium. A549 and H460 cells were incubated for 1.5 h and the SW900 cell line for 3 h at 37 °C. Next, 100 μL/well of lysis buffer (10% SDS and 10 mM HCl) was added. After 24 h, absorbance was read at 540 nm in an Infinite F200 spectrophotometer (TECAN, Männedorf, Switzerland).

To determine cytotoxicity in cells, lactate dehydrogenase (LDH) assays were performed 48 h after cell transfection with aptamers. Supernatants were collected and the Cytotoxicity Detection kit (LDH) (Roche, Madrid, Spain) reagent was added according to the manufacturer’s instructions. After incubation for 30 min at room temperature, the reaction was stopped by adding 1M HCl and absorbance was read at 490 nm in an Infinite F200 spectrophotometer (TECAN, Männedorf, Switzerland). To calculate the percentage of cytotoxicity, two controls were used: the supernatant of untreated cells (LDHlow) and of lysed cells (LDHhigh) with 0.2% Tritón X-100. Results are expressed as:Cytotoxicity (%)=LDHtreated−LDHlowLDHhigh−LDHlow×100

### 2.6. Trypan Blue Exclusion Test of Cell Viability

In order to determine the viable cells after transfection with apMNKQ2, cells were collected from the plates and 10 μL of cells were mixed with 10 μL of trypan blue and were counted on a TC20 counter (BioRad, Hercules, CA, USA). Those cells that excluded the dye were considered viable.

### 2.7. Cell Cycle Assay

Cells were plated at 5 × 10^5^ cells/well in a 6-well plate and transfected with the aptamers. Cells were trypsinized 24 h post-transfection, and both attached and floating cells were collected, fixed with PFA 4%, washed twice with PBS, and stored at 4 °C. Next, cells were incubated with FxCycle Violet Ready Flow Reagent (Thermo Fisher Scientific, Waltham, MA, USA) for 30 min at 25 °C and DNA content was evaluated with a 4-laser Attune NxT Acoustic Cytometer (Thermo Fisher Scientific, Waltham, MA, USA). Results were analyzed using FCSalyzer 0.9.22 alpha Software.

### 2.8. Apoptosis Assay

Cells were plated at 5 × 10^5^ cells/well in a 6-well plate, transfected with the aptamers during 24 h, resuspended, and stained with Annexin V Apoptosis detection kit (Canvax, Córdoba, Spain) as described previously [37] and analyzed with a 4-laser Attune NxT Acoustic Cytometer (Thermo Fisher Scientific, Waltham, MA, USA). Results were analyzed using FCSalyzer 0.9.22 alpha Software.

### 2.9. Clonogenic Assays

Cells were seeded at 5 × 10^4^ cells/well in 24-well plates and 16–24 h later, they were transfected with aptamers. After 16–24 h, cells were collected and seeded at 10^3^ cells/well in 6-well plates. Cells were incubated at 37 °C in a humidified incubator with 5% CO_2_ during 5–8 days. Then, cells were fixed with 1 mL/well of ethanol for 10 min at room temperature and stained with Giemsa 0.2% (Sigma, St. Louis, MO, USA) for 30 min. Remains of Giemsa were removed with water, images of colonies were taken and colonies were counted using ImageJ Java 1.8.0_172 software (National Institutes of Health, Bethesda, MD, USA).

### 2.10. Migration Assays

Migration assays were performed using transwell insert chambers (Corning, New York, NY, USA). Cells were seeded at 5 × 10^4^ cells/well in 24-well plates and transfected with aptamers 16–24 h later. After 16 h of serum-deprivation, cells were collected and added at 4 × 10^4^ cells/well into the upper chamber in 300 μL of serum-free medium. In the lower chamber, 500 μL of complete medium (10% FSB) was added as a chemoattractant. After 24 h, both media were removed and cells in the insert were fixed for 2 min with 4% formaldehyde followed by 20 min with 100% methanol. Next, cells were stained with 30 μM Hoechst 33342 for 15 min and washed twice with PBS. At least five photographs were taken for each sample using a fluorescence microscope Olympus IX70 (Olympus Iberia, Barcelona, Sapin). Cells were counted using ImageJ Java 1.8.0_172 software (National Institutes of Health, Bethesda, MD, USA).

### 2.11. Cell Adhesion Assays

Plates were coated with 10 μg/mg type I collagen (Sigma, St. Louis, MO, USA) or BSA (Sigma, St. Louis, MO, USA) as a control for 1 h at 37 °C. Cells were seeded at 5 × 10^4^ cells/well in 6-well plates and transfected with aptamers 16–24 h later. After 24 h, cells were collected, added to coated plates at 3 × 10^4^ cells/well in serum-free medium, and allowed to bind at 37 °C for 1 h. Wells were washed twice with PBS to remove non-adhered cells and an MTT assay was performed with 1 h of incubation between the addition of the MTT reagent and lysis buffer. After 24 h, absorbance was read at 540 nm in an Infinite F200 spectrophotometer (TECAN, Männedorf, Switzerland).

### 2.12. Zymography

The MMP activity of cells was analyzed through zymography. Cells were transfected and after 18 h of serum-deprivation, supernatants were collected and concentrated using Centricon tubes with a 3 KDa cutoff (Merck, Rahway, NJ, USA). Samples were mixed with loading buffer without mercaptoethanol and then separated in 7.5% polyacrylamide gels with 1 mg/mL gelatin. Gels were then incubated for 30 min in 2.5% Triton X-100 at room temperature and overnight at 37 °C in developing buffer (50 mM Tris-HCl pH 7.5, 200 mM NaCl, 5 mM CaCl_2_, and 0.02% Tween). Coomassie blue staining revealed the presence of MMP activity as clear bands against the blue background. Bands were quantified with ImageLab 6.1 software (BioRad, Hercules, CA, USA).

### 2.13. Aptamer Quantification

To quantify intracellular aptamers, cells were collected 4-, 24- and 48-h post-transfection, washed twice in PBS, lysed in 1 mL of H_2_O for 5 min, vortexed, and boiled at 90 °C for 10 min. Lysates were centrifuged at 12,000× *g* for 10 min and the aptamer in the supernatant was quantified by quantitative PCR. Pellets were resuspended in 10 μL of 0.1 M NaOH to determine protein concentration by BCA as described above. Results are expressed in fmol of aptamer/μg of protein. To quantify aptamer in tumors and organs, 50–100 mg of tissue were lysed and homogenized in NucleoZOL reagent (Macherey-Nagel, Düren, Germany) following the manufacturer’s instructions to isolate small RNA fraction, in which the aptamer is found. Results are expressed in fmol of aptamer/μg of tissue. Since apMNKQ2 is only 29 nucleotides in length, we designed a method that allows us to amplify this small-size aptamer in which apMNKQ2 acts as a primer on a template oligo-nucleotide called QR short (see Appendix A). After this reaction, both apMNKQ2 and QR short are elongated generating a 76-nucleotide template that can then be amplified with the appropriate primers (R3 and QF) (see Appendix A). Thus, in the absence of apMNKQ2, no amplification occurs, being the amplification proportional to the amount of apMNKQ2 present in the sample. The qPCR was performed using the AceQ qPCR SYBR^®^ Green Master Mix-Vazyme (Quimigen, Madrid, Spain) according to the manufacturer‘s instructions in a StepOne Plus Real-Time PCR system (Applied Biosystem, Waltham, MA, USA). The reaction mixture consisted of 1× Mix FastGene^®^ IC Green, 1 pmol QR short, 100 nM QF and R3 primers, and 1 μL of samples in a 10 μL/tube final volume. Aptamers were quantified using a standard curve (100 fmol–100 amol).

### 2.14. Quantification of mRNA

Total RNA was obtained as described above and was used to synthesize first strand cDNA using the SensiFAST^TM^ cDNA Synthesis Kit (Bioline, Segovia, Spain) following the manufacturer’s instructions. Products were used for qPCR amplification using the AceQ qPCR SYBR^®^ Green Master Mix-Vazyme (Quimigen, Madrid, Spain) according to the manufacturer’s protocol in a StepOne Plus Real-Time PCR system (Applied Biosystem, Waltham, MA, USA). Triplicate reactions of the targets and housekeeping genes were performed simultaneously for each cDNA template. Sequences of oligonucleotides are in Appendix A. To calculate the relative expression of each target gene, the 2^−ΔΔCt^ method was performed, using β-actin as a housekeeping normalization gene, where ΔCt is Ct_target_-C_thousekeeping_.

### 2.15. Tolerability Study

Non-tumor-bearing CD1 mice were used to determine the maximum tolerated dose (MTD) of apMNKQ2. The aptamer was administered once per day intraperitoneally (i.p.) for 7 consecutive days. We applied a modified “3 + 3” study design [38,39], using cohorts of 3 animals per dose and with the first cohort treated at a starting dose (1 mg/Kg). Subsequent cohorts were treated with increasing or decreasing doses according to the observed dose-limiting toxicities (DLTs) responses with guided subsequent doses, as described in [40]. DLT endpoints included weight loss (>20%), abnormal behavior, signs of physical discomfort, and/or death. Mice were evaluated daily and if any of the endpoints were met, animals were euthanized.

### 2.16. In Vivo Efficacy Assays

For in vivo experiments, mice were housed according to institutional guidelines, and all experimental procedures were performed in compliance with the institutional guidelines for the welfare of experimental animals approved by the Universidad Autónoma de Madrid Ethics Committee (CEI 60-1057-A068 and CEI 103-1958-A337) and La Comunidad de Madrid (PROEX 294/19) and in accordance with the guidelines for Ethical Conduct in the Care and Use of Animals as stated in The International Guiding Principles for Biomedical Research involving Animals, developed by the Council for International Organizations of Medical Sciences (CIOMS). Briefly, mice were housed according to the following guidelines: a 12-h light/12-h dark cycle, with no access during the dark cycle; temperatures of 65–75 °F (~18–23 °C) with 40–60% humidity; a standard diet with fat content ranging from 4–11%; sterilized water was accessible at all times; for handling, mice were manipulated gently and as little as possible; noises, vibrations, and odors were minimized to prevent stress and decreased breeding performance; and enrichment was always used per the facility’s guidelines to help alleviate stress.

Female 9-week-old Foxn1^nu/nu*^ (Janvier Labs, Le Genest-Saint-Isle, France) were injected subcutaneously in the dorsal flanks with 8 × 10^6^ A549 cells resuspended in 100 μL Matrigel (Corning, New York, NY, USA) per injection. Once tumors were established, mice were randomized into 3 groups (5 mice per group) and injected intraperitoneally with vehicle control (selection buffer), 10 mg/kg apMNKQ2 or 25 mg/kg apMNKQ2. Aptamer injections were daily and tumor volumes were determined twice per week for 25 days using a manual caliper. At the time of sacrifice, tumors were excised and weighed. One half was frozen in liquid nitrogen for western blot and qPCR analysis and the other half was fixed in 4% PFA and processed for histologic analysis. At the end of the treatment, the TGI ratio (%) was calculated using the following formula: TGI (%) = [1 − (Volume of the treated group)/(volume of the control group)] × 100 (%).

### 2.17. Statistical Analyses

Results are presented as means ± standard error of the mean (SEM) unless stated otherwise. Pair-wise multiple comparisons were performed with one-way ANOVA (two-sided) with Turkey’s test adjustment, as indicated in the figure legends. A student’s t-test was used to determine differences between the means of groups. The *p*-values < 0.05 were considered statistically significant. All analyses were performed using GraphPad Prism version 8.0 (San Diego, CA, USA).

## 3. Results

### 3.1. apMNKQ2 Inhibits Cell Viability in NSCLC Cell Lines

In our previous work [37], four sequences were designed (apMNKQ1, apMNKQ2, apMNKQ3, and apMNKQ4) from the aptamer apMNK2F against MNK1b and tested in breast cancer cell lines. Here, to study the effect of the four sequences in the context of lung cancer, A549 and SW900 cell lines were transfected with the four aptamers at 250 nM, and cell viability was assessed 48 h later by measuring MTT activity. The nonspecific aptamer (p29) was used as a control. Results (Appendix A) showed that apMNKQ2 had the greatest effect on MTT activity in both cell lines, producing a significant reduction in cell viability of 77% in A549 and 51% in SW900 cells.

Consequently, the effect of apMNKQ2 on cell proliferation and toxicity was evaluated in three cell lines (A549, SW900, and H460) representing the main types of non-small cell lung cancer (adenocarcinoma, squamous cell carcinoma, and large cell carcinoma, respectively). Cells were transfected with increasing concentrations (0–500 nM) of apMNKQ2 or the nonspecific control aptamer (p29). MTT activity was measured after 48 h and to study possible cell death by necrosis, LDH enzyme activity was simultaneously measured. Results showed that apMNKQ2 decreased MTT activity in all three cell lines in a concentration-dependent manner, with A549 cells being the most sensitive (Figure 1a). The nonspecific p29 aptamer had a slight effect on the cell lines (less than 20%). The IC50 values were 70 nM, 250 nM, and 100 nM for A549, SW900, and H460, respectively. This effect was not necrosis-mediated since there was little to no cytotoxicity (Figure 1b).

When aptamers are used against intracellular targets it is important to determine their half-life inside the cell, which may be affected by their molecular nature or structure. To study the intracellular stability of apMNKQ2, cells were transfected according to their IC50. After 4, 24, and 48 h, cells were lysed and apMNKQ2 was quantified by qPCR. As shown in Figure 1c, apMNKQ2 is stable in all three cell lines for at least 48 h.

### 3.2. apMNKQ2 Induces Apoptosis, Cell Cycle Arrest, and Inhibits Colony Formation in Lung Cancer Cells

To determine whether the decrease in cell viability was a consequence of apMNKQ2-induced apoptosis, we analyzed the apoptotic marker cleaved-PARP. Aptamer-transfected cells were lysed after 24 h of transfection (cell viability 24 h after transfection is shown in Appendix A) and analyzed by western blot as described in the Material and Methods section (Section 2). Figure 2a shows that the cleavage of PARP by caspase 3 occurs in the three cell lines albeit to a different extent. A decrease in the uncleaved form of PARP was observed in A549 and SW900 cell lines, being statistically significant in A549 cells. In addition, we observed a significant increase in both the percentage of Annexin-V positive and propidium iodide (PI) negative cells (Figure 2b) and the percentage of the sub-G1 phase in the A549 cells (Appendix A), indicating that apMNKQ2 induces apoptosis in this cell line.

We also analyzed XIAP and MCL1 levels since both are antiapoptotic proteins regulated by MNK1 [23,41,42]. Results show that apMNKQ2 significantly decreased both XIAP and MCL1 levels in A549 and SW900 cell lines, again confirming the proapoptotic effect of apMNKQ2 (Figure 2c,d). However, apMNKQ2 seemed to significantly increase the levels of MCL1 in H460 cells (Figure 2c).

Next, we determined the effect of the aptamers on cell cycle distribution. Cells were transfected and 24 h later analyzed by flow cytometry. Figure 3a shows that apMNKQ2 increased the percentage of cells in the G1 phase in A549 (6.6%) and H460 (7.5%) cells compared with control cells, indicating that apMNKQ2 induces an arrest in the G1 phase of the cell cycle. Surprisingly, apMNKQ2 induced an increase in the percentage of cells in the G2/M phase (3.8%) in SW900 cells.

Moreover, we studied the effect of apMNKQ2 on colony formation and observed that apMNKQ2 significantly reduced the clonogenic capacity of the three cell lines (Figure 3b). The unspecific p29 aptamer had a slight but significantly less effect compared to apMNKQ2.

### 3.3. apMNKQ2 Inhibits Migration, Invasion, Cell Adhesion, and Epithelial-Mesenchymal Transition (EMT) in Lung Cancer Cells

Metastatic dissemination requires cancer cells to detach from the primary tumor and colonize distant organs through pivotal steps such as cell migration, invasion, and adhesion. We have studied how apMNKQ2 affects all these cell features in order to determine its antimetastatic potential. We performed transwell migration assays to determine the migratory ability of lung cancer cells after being transfected with apMNKQ2. Results showed that the number of migrated cells was lower in the three cell lines following transfection with apMNKQ2, and statistically significant in both A549 and SW900 cells (Figure 4a).

The Invasiveness of cancer cell lines depends on the expression of proteins such as MMPs. Thus, we analyzed the invasive potential of the three cell lines by zymography in order to detect changes in the proteolytic activity of matrix metalloproteases MMP2 and MMP9 after transfection with apMNKQ2. As Figure 4b shows, apMNKQ2 decreased MMP9 and MMP2 activity in both A549 and SW900 cells, although the effect was only statistically significant for MMP9 in A549 cells. In H460 cells, however, apMNKQ2 seemed to increase MMP9 activity and did not produce changes in MMP2.

Next, we analyzed whether apMNKQ2 affected the ability of cells to adhere to extracellular matrix components such as type I collagen. For these assays, 24 h after transfection with aptamers, cells were reseeded in plates previously coated with type I collagen in order to measure adherence. Subsequently, MTT assays were performed to quantify the number of adherent cells. Results demonstrated that apMNKQ2 significantly reduced cell adhesion in A549 cells but did not have any effect on SW900 and H460 cells (Figure 4c).

Finally, we studied the effect of apMNKQ2 on epithelial-mesenchymal transition (EMT), in which cancer cells lose epithelial features and acquire a mesenchymal phenotype. For this purpose, cells were transfected with apMNKQ2 and the expression of the epithelial markers E-cadherin and occludin and the mesenchymal marker N-cadherin were analyzed by western blotting. E-cadherin and N-cadherin were only detected in A549 and SW900 cells while occludin was detected in the three cell lines (Appendix A). Results showed that apMNKQ2 significantly reduced E-cadherin levels in both A549 and SW900 cells (Figure 4d) and increased occludin levels in H460 cells (Figure 4e). Moreover, apMNKQ2 produced a significant reduction in N-cadherin expression levels in SW900 cells (Figure 4f).

### 3.4. Effect of apMNKQ2 on MNK1 Isoforms

The expression levels of both MNK1 isoforms were analyzed as well as eIF4E phosphorylation after cell transfection with aptamers. Figure 5a shows that apMNKQ2 increased eIF4E phosphorylation in A549 cells while no changes were observed in SW900 and H460 cells. MNK1a was significantly reduced in the three cell lines after apMNKQ2 transfection (Figure 5b), but MNK1b did not change in any of the three cell lines (Figure 5c). The reduction of MNK1a observed in SW900 cells may be a consequence of a reduction in mRNA levels (Appendix A).

### 3.5. apMNKQ2 Reduces Tumor Growth In Vivo

To investigate the antitumor efficacy of apMNKQ2, we first performed a tolerability study as described in the Materials and Methods section (Section 2). During the course of this experiment, no DLTs were observed at any of the doses tested, including the maximum feasible dose of 400 mg/kg. Thus, for practical purposes, doses of 10 and 25 mg/kg were used in the xenograft studies. Then, A549 cells were subcutaneously xenografted in athymic nude mice, and apMNKQ2 was administered intraperitoneally at 10 mg/kg and 25 mg/kg. Our preliminary results showed that treatment with apMNKQ2 produced a reduction in both tumor volume (Figure 6a) and weight (Figure 6b) reaching a tumor growth inhibition (TGI) of 18.2% and 28.5% for 10 mg/kg and 25 mg/kg of apMNKQ2, respectively. However, these results were not statistically significant, probably due to unexpectedly slow tumor growth and/or the necessity of a higher dose of the aptamer.

We analyzed aptamer uptake after intraperitoneal administration by qRT-PCR and demonstrate that apMNKQ2 reaches different organs, such as the lung or pancreas but not the brain and is cleared by the kidney and liver (Appendix A). In tumors, apMNKQ2 reached the tumor in a dose-dependent manner (Figure 6c) and exerted a biological effect on the tumor, with no toxic macroscopic effects observed in other organs.

In order to determine the association between the anti-tumor effects observed in vivo and the inhibition of MNK1 downstream signaling, XIAP and MCL1 levels were analyzed from tumor samples by western blotting. Results showed a reduction of both proteins in 25 mg/kg apMNKQ2-treated tumor samples (Figure 6d). These results corroborate our in vitro findings of the apoptotic effect of apMNKQ2 in A549 cells.

## 4. Discussion

Lung cancer is considered one of the most invasive cancers and the fastest to metastasize, being the leading cause of cancer-related death worldwide and the third most frequently diagnosed in 2022 after colorectal and breast cancer [2]. To date, therapeutic strategies to treat lung cancer, such as chemotherapy, molecular targeted therapy, or immunotherapy, have been developed [43] and have improved the survival of lung cancer patients. Despite these advances, the prognosis for patients with lung cancer remains poor, highlighting the need for new therapeutic approaches.

MNKs are involved in several types of cancer [25] including NSCLC, where MNK1 overexpression correlates with poor overall patient survival [44]. This suggests that blocking the MNK/eIF4E pathway may be a good strategy to treat NSCLC.

Here we studied the antitumor potential of apMNKQ2 as an antitumor agent in NSCLC using in vitro and in vivo assays. The three cell lines used harbor KRAS mutations, SW900 cells carry an inactivated mutation in the tumor suppressor gene TP53, and H460 cells also harbor PI3KCA mutations [45,46].

One of the most important characteristics of aptamers is their stability and half-life inside cells, and apMNKQ2 is stable inside lung cancer cells for at least 48 h, which may be an advantage for its clinical application in terms of decreasing the number of injections and improving patient comfort [47].

Inhibiting tumorigenic characteristics is an important aspect of the development of antitumor drugs. Consequently, we show that apMNKQ2 reduces the viability in A549 and SW900 cells via apoptosis induction, but to a less extent in the latter. While SW900 cells are not Annexin-V positive after transfection with apMNKQ2, PARP was cleaved. In addition, XIAP and MCL1 are antiapoptotic proteins that are overexpressed in several types of cancers and whose expression is regulated by MNK1 [23,41,42]. Moreover, MCL1 overexpression inhibits apoptosis and promotes cell survival in NSCLC [48]. The apMNKQ2 reduced both XIAP and MCL1 in A549 and SW900 cells both in vitro and in vivo (the latter only for A549) proving its effect through MNK1, and confirming the pro-apoptotic effect of the aptamer. However, despite the fact that other MNK inhibitors [26] promote the induction of apoptosis in H460 cells, apMNKQ2 did not induce it, which may explain the lack of effect on MCL1 and XIAP levels.

Some MNK inhibitors produce a cell cycle arrest in breast cancer cells [19,49]. For example, the MNK1 inhibitor BAY1143269 induces a G0/G1 arrest in NSCLC cells. This is consistent with our results in A549 and H640 cells in which a G1 phase arrest is produced when cells are transfected with apMNKQ2. Surprisingly, the aptamer induces a G2 phase arrest in SW900 cells. This difference among the three cell lines could be explained since both A549 and H460 cells contain a wild-type TP53, while SW900 cells carry an inactivated mutation in TP53.

MNKs also play an important role in metastasis [16], and several MNK inhibitors reduce cell migration and invasion [26,50,51,52,53]. Indeed, apMNKQ2 decreased migration and invasion of NSCLC cells. During metastasis, cells migrate and attach to extracellular matrix (ECM) components. apMNKQ2 was also able to reduce cell adhesion capacity in A549 cells. β1 integrin is involved in the adhesion of cells to the surroundings ECM and its inhibition reduces the adhesion of A549 cells by inhibiting the ERK1/2 signaling pathway [54]. This suggests that apMNKQ2 may reduce cell adhesion through MNK1 inhibition. The catalytic subunit p110α mediates β1 integrin-regulated activation of AKT [55], which could contribute to the tumorigenic properties of cells expressing constitutively active p110 α, as in H460 cells.

Similarly, apMNKQ2 alters cell migration and MMP activity only in A549 and SW900 cells. The PI3K/AKT/mTOR pathway plays an important role in lung cancer cell migration and invasion, specifically in the regulation of MMP2 and MMP9 [56]. These data suggest that the increased activity in the PI3K pathway in H460 cells could compensate for the effects of apMNKQ2.

Therefore, apMNKQ2 seems to exert more potent effects in A549 and SW900 cells than in H460 cells probably due to the PI3KCA mutation in the latter. It would be interesting to use apMNKQ2 in combination with inhibitors of the PI3K pathway to increase the sensitivity of H460 cells to apMNKQ2.

apMNKQ2 significantly reduced N-cadherin levels in SW900 cells, which is in accordance with other MNK inhibitors such as VNLG-152 that reverse EMT in prostate cancer [57]. The aptamer also increased the epithelial marker occludin in H460 cells, recovering part of the epithelial characteristics in these cells. Since Snail is a direct repressor of E-cadherin [58], the overexpression of Snail in A549 cells (data not shown) may be associated with low E-cadherin expression in these cells.

Some MNK inhibitors such as MNK-I1, BAY1143269, MNK-7g, eFT508 or novel retinamides inhibits eIF4E phosphorylation [26,51,59,60,61]. apMNKQ2 also inhibits eIF4E phosphorylation in breast cancer cells [37]; however, in NSCLC cells, apMNKQ2 did not affect eIF4E phosphorylation, which is in accordance with other MNK inhibitors such as apMNK2F and apMNK3R aptamers [36] or ferrocene analogs [62]. Since eIF4E phosphorylation is mainly mediated by MNK2 in breast cancer cells MDA-MB-231 [36], this could also occur in lung cancer cells. Moreover, MNK2b acts as a proto-oncogene increasing eIF4E phosphorylation [63] and in NSCLC, MNK2 is involved in tumorigenesis through eIF4E phosphorylation [21]. To answer this unknown, we propose to use of CRISPR/Cas, which is currently being developed in the laboratory.

Even though eIF4E is the best characterized substrate of MNK1, it is possible that eIF4E-independent effects, through other MNK substrates, may contribute to the observed MNK-mediated effects of apMNKQ2. Here we show that apMNKQ2 exerts its effects through proteins regulated by MNK1 such as XIAP and MCL1, but in the last few years, other proteins regulated by MNK1 have been described such as NODAL [64], NDRG1 [65] or ANGPTL-4 [66], which may be involved in the antitumoral mechanism of apMNKQ2.

We have studied different administration routes of apMNKQ2 in preliminary assays in mice, such as oral, intravenous, and intraperitoneal routes. The latter was the most effective in breast and pancreatic cancer tumor models and was therefore chosen for the in vivo experiments presented herein. We have shown that there is a downward trend in tumor growth after treatment with apMNKQ2. Although tumor growth with xenografted A549 cells was slower than expected, we have obtained promising results that can be improved by increasing the dose of the aptamer. It must be emphasized that in previous studies [37] we used a transfectant as a vehicle to administer apMNKQ2 into animals, while in this study apMNKQ2 was injected without the transfectant, which is an advantage since it allowed us to increase the aptamer dose. Importantly, in the absence of the transfectant, the aptamer still reached the tumor and was easily detectable by qRT-PCR and increased as a function of the dose. Indeed, further studies are still necessary to determine the antitumor mechanism and the full antitumor potential of apMNKQ2. Thus, apMNKQ2 is currently being tested in a patient-derived xenograft (PDX) of NSCLC. This will allow us to learn more about the therapeutic potential of the aptamer in a more physiologically relevant tumor model.

One of the most important points to take into account is the toxicity commonly observed with antitumor chemotherapeutics [67] or immunotherapy and molecular targeted therapy, even though the latter two are better tolerated, side effects can occur [68,69]. Toxicity regarding aptamers is very limited. For example, the aptamers NOX-A12 or ApTOLL have been administered to patients with no signs of toxicity [70]. Moreover, treatment with apMNKQ2 was well tolerated in breast cancer in vivo assays as indicated by no difference in body weight [37] indicating the low toxicity of apMNKQ2. Likewise, daily administration of apMNKQ2 in this study showed no obvious signs of toxicity in mice. Moreover, we performed a maximum tolerated dose (MTD) assay in CD1 mice and no toxicity was observed at any dose, including the highest dose tested (400 mg/kg, maximum feasible dose), indicating that the data obtained so far demonstrates low toxicity for apMNKQ2. However, future ADME Tox are still necessary and are currently planned.

In summary, we show that apMNKQ2 inhibits the tumorigenic and metastatic processes of lung cancer cells, and preliminary in vivo experiments indicate that apMNKQ2 can reduce tumor growth and induce apoptosis in an adenocarcinoma xenograft model. Further studies are still needed in order to elucidate the mechanism of action of apMNKQ2 as well as its pharmacokinetics and pharmacological security to expedite its possible use as a therapeutic tool in the different types of cancers where MNK1 plays an important biological role.

## 5. Patents

C.P-D, V.M.G. and M.E.M. declare that a patent application has been filed relating to this work.

## Figures and Tables

**Figure 1 pharmaceutics-15-01273-f001:**
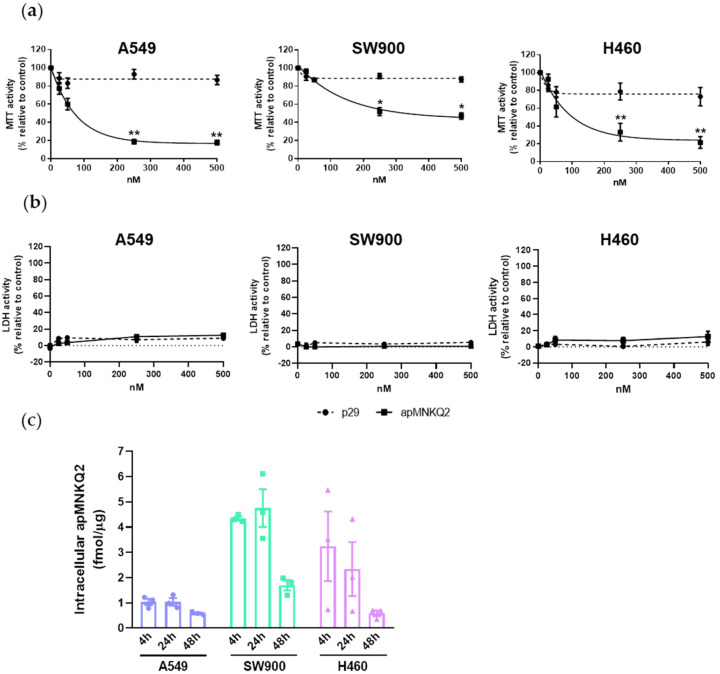
Aptamer effect on the viability of lung cancer cells. (**a**) Effect of apMNKQ2 on MTT and (**b**) LDH activity. A549 and H460 cells were seeded at 6 × 10^3^ cells/well and SW900 cells were seeded at 10^4^ cells/well in 96-well plates. After 48 h MTT and LDH assays were performed. Graphs represent the mean ± SEM of 6 and 3 independent experiments. * *p* < 0.05 and ** *p* < 0.01 relative to p29 control. (**c**) The half-life of apMNKQ2 in lung cancer cells. Cells were seeded at 5 × 10^5^ cells/well in 6-well plates and transfected with the corresponding IC50 of apMNKQ2 16–24 h later. After 4, 24, and 48 h, cells were lysed and apMNKQ2 was quantified through qPCR. Results are expressed as fmol of aptamer/μg of protein. Bars represent mean ± SEM of 3 independent experiments. Solid blue circles (A549 cell line), green squares (SW900 cell line) and violet triangles (H460 cell line) represent one individual value.

**Figure 2 pharmaceutics-15-01273-f002:**
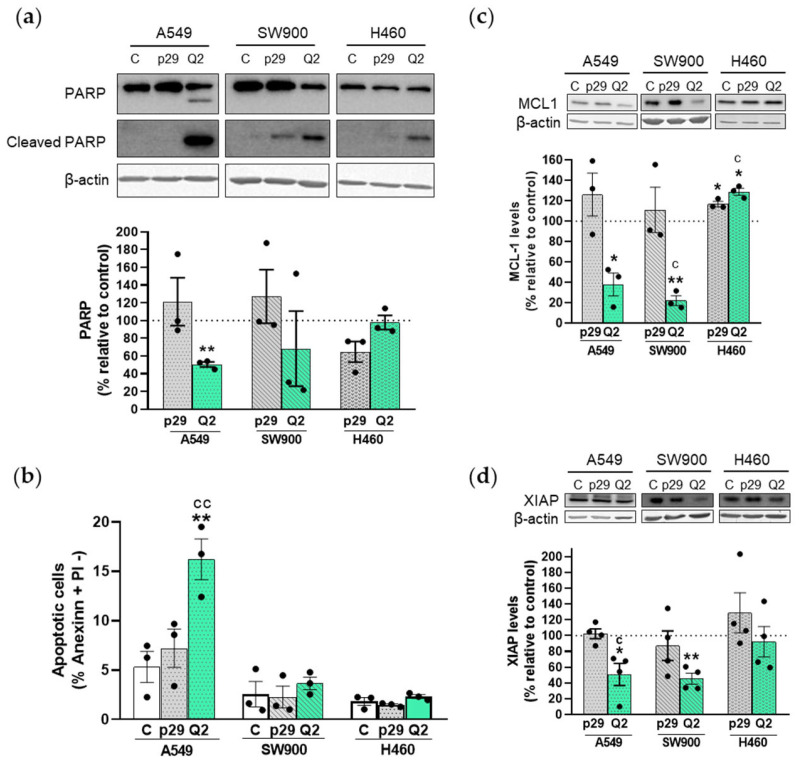
Apoptotic activity of apMNKQ2 on NSCLC cells. (**a**) On the top, lysates (20 μg) were resolved in 10% SDS-PAGE gels for PARP and cleaved PARP. On the bottom, densitometry analyses of total PARP in the three cell lines. Data were normalized to respective β-actin bands and expressed as the percentage relative to control (C) cells. The graphs represent the mean ± SEM of 3 independent experiments. (**b**) Quantification of early apoptotic cells by Annexin-V staining. The bars represent the mean percentage of Annexin-V positive and PI negative cells ± SEM of 3 independent experiments. (**c**,**d**) Densitometry analyses of XIAP and MCL-1. Both were normalized to respective β-actin bands and expressed as the percentage relative to control (C) cells. The graphs represent the mean ± SEM of 3–4 independent experiments. Lysates (20 μg) were resolved in 12% gels for XIAP and MCL-1 and western blot analysis was carried out using specific antibodies (see Appendix A). A representative blot is shown on the top of each graph. * *p* < 0.05 and ** *p* < 0.01 relative to control cells; ^c^ *p* < 0.05 and ^cc^ *p* < 0.01 relative to p29 control. Control is shown as C, p29 is the unspecific aptamer used as a negative control, and apMNKQ2 is shown as Q2. Each solid black circle represents one individual value.

**Figure 3 pharmaceutics-15-01273-f003:**
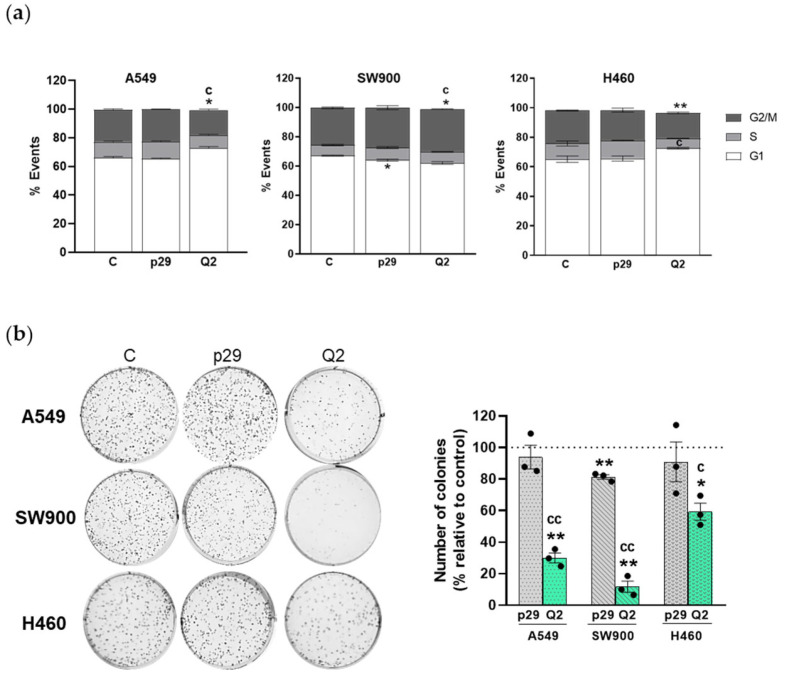
apMNKQ2 arrests cell cycle progression and colony formation in NSCLC cells. (**a**) Cells were transfected with apMNKQ2 at IC50 concentrations. After 24 h, cells were stained with PI and analyzed by flow cytometry. The bars represent the mean percentage of cells gated in each phase, G1 (white), S (grey), and G2/M (dark grey) ± SEM of 3 independent experiments. (**b**) Cells were transfected with aptamers at IC50 and after 24 h reseeded at 10^3^ cells/well in 6-well plates. After 7–8 days, colonies were fixed, stained, and counted. (Left) shown are representative images. (Right) The bars represent the mean percentage of the number of colonies obtained ± SEM of 3 independent experiments relative to control (C). * *p* < 0.05 and ** *p* < 0.01 relative to control cells; ^c^ *p* < 0.05 and ^cc^ *p* < 0.01 relative to p29 control. Control is shown as C, p29 is the unspecific aptamer used as a negative control, and apMNKQ2 is shown as Q2. Each solid black circle represents one individual value.

**Figure 4 pharmaceutics-15-01273-f004:**
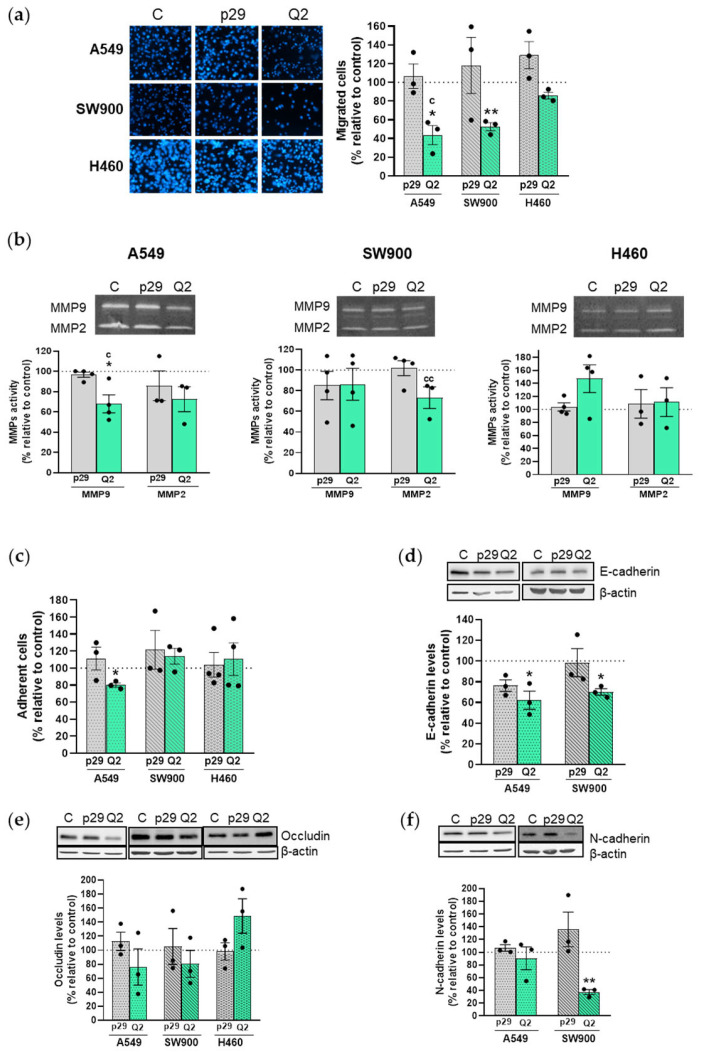
ApMNKQ2-mediated effects on migration, invasion, and EMT in NSCLC cells. (**a**) Cells were transfected with aptamers at IC50, maintained in serum-deprived medium for 16 h, and collected and reseeded at 4 × 10^4^ cells/well in the upper chamber of a transwell. Representative images are shown in the left panel. The graphs represent the mean percentage of migrated cells ± SEM relative to the value of control (C) cells from 3 independent experiments. (**b**) Zymography was performed as described in Materials and Methods. The graphs represent the mean percentage of MMP activity ± SEM relative to the value of control (C) cells from 3–4 independent experiments. (**c**) Cells were transfected with aptamers at IC50 and reseeded in plates previously coated with type I collagen. After binding, cells were quantified by MTT assay. The graphs represent the mean percentage of adherent cells ± SEM relative to the value of control cells from 3 independent experiments. (**d**–**f**) Western blot and corresponding densitometry analyses of lysates (20 μg) that were subjected to SDS-PAGE (10%). Specific antibodies are shown in Appendix A. Actin detection was used as a loading control. A representative blot is shown at the top of each graph. Densitometric quantification of E-cadherin, occludin, and N-cadherin was normalized to respective β-actin bands, and levels are expressed as a percentage relative to control cells. The graphs represent mean percentage levels ± SEM of 3 independent experiments. * *p* < 0.05 and ** *p* < 0.01 relative to control cells; ^c^ *p* < 0.05 and ^cc^ *p* < 0.01 relative to p29 control. Control is shown as C, p29 is the unspecific aptamer used as a negative control, and apMNKQ2 is shown as Q2. Each solid black circle represents one individual value.

**Figure 5 pharmaceutics-15-01273-f005:**
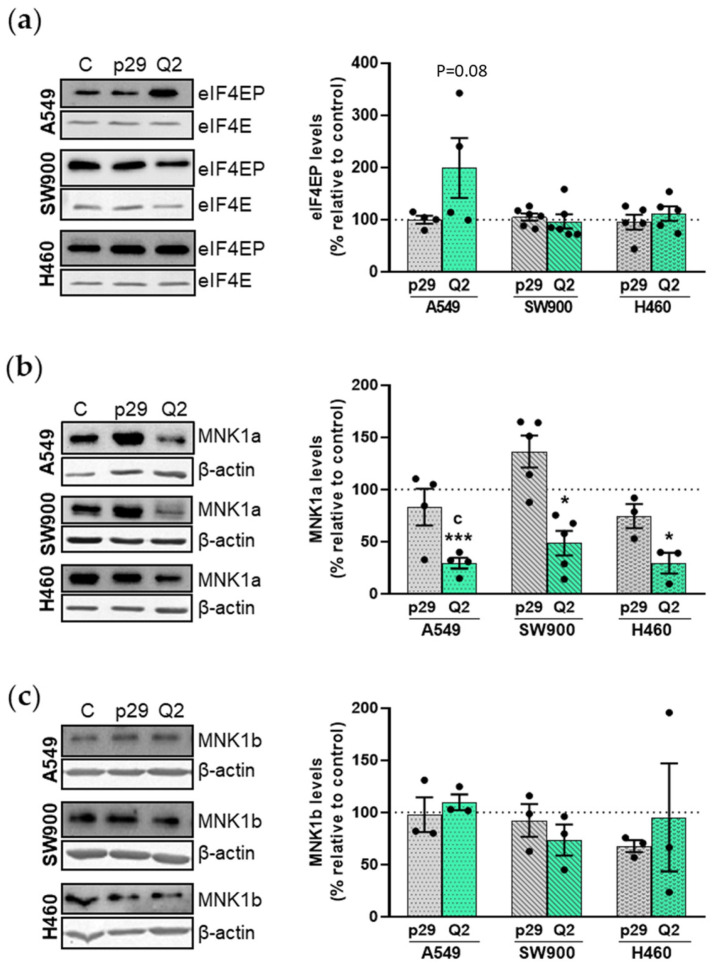
Effect of apMNKQ2 on eIF4E phosphorylation and MNK1 isoforms. Lysates (20 μg) were subjected to SDS-PAGE (12%) and western blotting was performed using specific antibodies (see Appendix A). Actin detection was used as a loading control. A representative blot and corresponding densitometry analyses are shown in each figure. (**a**) Quantification of eIF4EP was normalized to total eIF4E levels and expressed as a percentage relative to control (C) cells (**b**) and (**c**) quantification of MNK1a/b was normalized to β-actin levels and expressed as a percentage relative to control cells. The graphs represent mean values ± SEM of 3–5 independent experiments. * *p* < 0.05 and *** *p* < 0.001 relative to control cells; ^c^ *p* < 0.05 relative to p29 control. Control is shown as C, p29 is the unspecific aptamer used as a negative control, and apMNKQ2 is shown as Q2. Each solid black circle represents one individual value.

**Figure 6 pharmaceutics-15-01273-f006:**
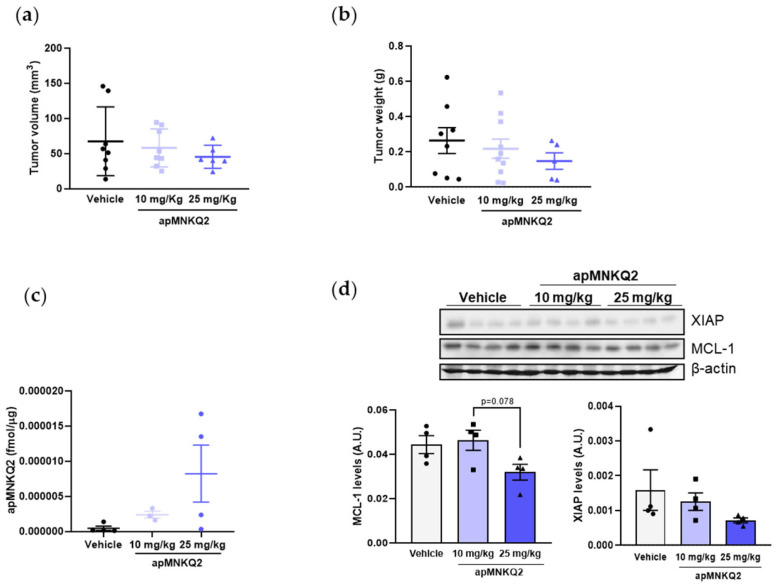
apMNKQ2 efficacy in an adenocarcinoma xenograft model. (**a**,**b**) Reduction in (**a**) volume and (**b**) weight of A549 xenografts following treatment with apMNKQ2 at 10 and 25 mg/kg daily. Mice were treated intraperitoneally with apMNKQ2 or vehicle control (selection buffer) for 4 weeks. Scatter plots represent the mean ± SEM from 6-8 tumors. Solid black circles (vehicle), blue squares (apMNKQ2 at 10 mg/Kg) or blue triangles (apMNKQ2 at 25 mg/Kg) represent one individual value. (**c**) Quantification of apMNKQ2 levels in tumors by qRT-PCR. RNA fraction, where apMNKQ2 is found, was obtained from tumor samples and used in qPCR analyses as described in Material and Methods. Results are expressed as pmol/μg of tissue. Scatter plot represents the mean ± SEM from 3–4 tumors. Each solid circle represents one individual value. (**d**) Proteins were obtained from tumor samples as described in Material and Methods and lysates (20 μg) were subjected to SDS-PAGE (12%). Western blotting was performed using specific antibodies (see Appendix A). Actin detection was used as a loading control. Blots are shown on the top of the graphs. The bars represent mean protein levels ± SEM, expressed as arbitrary units (A.U.) for 4 tumors from each treatment arm. Each solid black symbol represents one individual value.

## Data Availability

Not applicable.

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
