# Peer review of "An Aptamer against MNK1 for Non-Small Cell Lung Cancer Treatment"

_pharmaceutics, 2023, doi:10.3390/pharmaceutics15041273_

Round 1

Reviewer 1 Report (Previous Reviewer 1)

Dr Elena Martin's work is very interesting and shows consistent results. The responses from my 1st review  were satisfactory and now allow for a better understanding of the article. There are still last questions I would like to clarify:

1. How does the internalization of aptamers occur in tumor cells in vivo? In cell culture experiments, aptamers were transfected to verify the intracellular effect. Why were aptamers transfected into cell cultures and not just incubated in culture medium? Did the authors test the simple incubation of aptamers with cell culture? Does internalization and target blocking effect occur under these conditions?

2. Do aptamers bind to any extracellular receptor or carrier protein that will mediate internalization? Is it possible to demonstrate this in an in vitro assay?

3. The authors show an experiment to evaluate the intracellular stability of aptamers in cell culture. In the material and methods, it is mentioned that aptamers do not have modified bases. Could protection of aptamers with modified bases improve stability and biological effect? What is the advantage of using unmodified bases?

Author Response

  1. How does the internalization of aptamers occur in tumor cells in vivo? In cell culture experiments, aptamers were transfected to verify the intracellular effect. Why were aptamers transfected into cell cultures and not just incubated in culture medium? Did the authors test the simple incubation of aptamers with cell culture? Does internalization and target blocking effect occur under these conditions?

This is a very interesting question that we are currently investigating. Indeed, we checked the simple incubation of apMNKQ2 in cell culture without lipofectamine or any other vehicle, observing the entry of apMNKQ2 into cells, but to a lesser extent than when the aptamer is transfected with lipofectamine. Moreover, simple incubation of apMNKQ2 in cell culture had no effect on cell viability, so for this reason lipofectamine was used in in vitro assays.

We hypothesize that when the aptamer enters into cells without vehicle, the entry pathway is different than when lipofectamine is used. Perhaps a different entry mechanism results in a pH change or other event that may alter the structure and/or or stability of the aptamer, leading to a lack of effect of the aptamer in cell culture.    

  1. Do aptamers bind to any extracellular receptor or carrier protein that will mediate internalization? Is it possible to demonstrate this in an in vitro assay?

As we mention above, we do not know the mechanism of aptamer internalization when lipofectamine is not used in an in vitro assay. We have tried to demonstrated it using EEA1 and LAMP-1 antibodies, in order to mark the endosome pathway at various stages, and apMNKQ2-Alexa4888 with and without lipofectamine in immunocytochemistry at times 1, 2, 4 and 24 hours. Even so, we have not been able to distinguish the mechanism of internalization.

In addition, it is a very interesting point in terms of in vivo assays. As we mention in the discussion section, the aptamer still reaches the tumor in the absence of a transfectant but further studies are necessary to determine the mechanisms by which this occurs. One of our hypotheses is that the aptamer could reach the tumor protected by plasma proteins such as albumin, and we are also trying to elucidate this, taking into account that there may be a multitude of macromolecular interactions in vivo that do not occur in the in vitro assays.

  1. The authors show an experiment to evaluate the intracellular stability of aptamers in cell culture. In the material and methods, it is mentioned that aptamers do not have modified bases. Could protection of aptamers with modified bases improve stability and biological effect? What is the advantage of using unmodified bases?

Effectively, several chemical modifications have been developed to improve the pharmacokinetic properties of aptamers used in therapy. Sugar ring modifications are made for impairing nuclease-mediated degradation and the addition of a bulky moiety such as cholesterol or PEG is carried out to overcome renal filtration. Chemical modifications may be problematic since unnatural nucleotides could cause toxic effects. Moreover, the synthesis of unmodified aptamers is cheaper than aptamers with modified bases. In this regard, one of the main advantages of the apMNKQ2 aptamer is its high resistance to nucleases, which allows its use without any modification.

Reviewer 2 Report (New Reviewer)

  Rebeca Carrión-Marchante  et al evaluated the effect of the anti- MNK1 apMNKQ2 aptamer in lung cancer monitoring viability, proliferation  and migration in three NSCLC cells and evaluating its in vivo efficacy. The study is interesting. However, several points should be addressed prior to further consideration.

A general concern arises from the different behavior of the aptamer in the three cell lines that needs to be discussed.

In the discussion section, the authors state that since “CGP57380 and BAY1143269 treatment reduce the pro-survival factor survivin in H460 cells, apMNKQ2 may be expected to induce apoptosis in H460 cells independent of XIAP or MCL1”. However, it seems that apart from the involved signaling the aptamer has different effects on this cell line as well as on SW900, as compared to A549 cells.

 Specific comments:

-       -  In figure 2, while the induction of apoptosis is clear for A549, it is less evident (and not very significant) in the other cell lines especially H460. How the authors explain this difference? To better analyze apoptosis, authors need to include additional assays (Anexin/PI) to monitor the % of apoptotic cells and caspase activity.

-       - In Figure 2d, at difference of MTT the efficacy on colony formation seems to be better in SW900. Authors should comment this difference.

-        - Similarly, in Figure 3, the aptamer alters cell migration only in A549 and SW900 and impairs MMPs only in A549. How the authors explain the difference between the three cells?

-    - In addition to cell migration assay, it would be useful to test aptamer ability to suppress cell invasion (can be done with Matrigel-coated transwell chambers).

-       - Why E-cadherin N-cadherin were only detected in A549 and SW900 cells? These markers should be expressed also by H460. I suggest the authors to add additional analyses (i.e RTqPCR) and/or include more EMT markers.

-       - The authors “do not rule out the possibility that apMNKQ2 may inhibit proliferation of lung cancer cells via cell cycle arrest”, this can be easily checked by FACS analyses. I suggest including this analyses

Minor:

-       -    Figure 4a, 5a and b lack of statistical analyses. Please add significance

-     I suggest the authors check for few text errors. 

For example, Page 6 line 291: “In our previous work [37], four sequences were designed (apMNKQ1, apMNKQ2, 291 apMNKQ3 and apMNKQ4) from the aptamer apMNK2F against MNK1b and tested in breast cancer cell lines”

-         - It is unclear to me why some text part are in yellow, is it a revised manuscript?

Author Response

   Rebeca Carrión-Marchante  et al evaluated the effect of the anti- MNK1 apMNKQ2 aptamer in lung cancer monitoring viability, proliferation  and migration in three NSCLC cells and evaluating its in vivo efficacy. The study is interesting. However, several points should be addressed prior to further consideration.

A general concern arises from the different behavior of the aptamer in the three cell lines that needs to be discussed.

In the discussion section, the authors state that since “CGP57380 and BAY1143269 treatment reduce the pro-survival factor survivin in H460 cells, apMNKQ2 may be expected to induce apoptosis in H460 cells independent of XIAP or MCL1”. However, it seems that apart from the involved signaling the aptamer has different effects on this cell line as well as on SW900, as compared to A549 cells.

As detailed below, we have discussed, in the discussion section (Highlighted in the new version of the manuscript,) the different behavior of apMNKQ2 in the three lines relative to apoptosis, cell cycle and migration.

 Specific comment

-      - In figure 2, while the induction of apoptosis is clear for A549, it is less evident (and not very significant) in the other cell lines especially H460. How the authors explain this difference? To better analyze apoptosis, authors need to include additional assays (Anexin/PI) to monitor the % of apoptotic cells and caspase activity.

We have carried out Annexin-V assays and we have added them to the 3.2 results section and in the new Figure 2b.

apMNKQ2 induces apoptosis in both A549 and SW900 cells (detection of cleaved PARP and reduction of total PARP), although the apoptosis induction is only significant and clear in A549 cells via all techniques used (cleaved and total PARP, MCL-1, XIAP, Annexin-V and subG1 phase). Indeed, apMNKQ2 does not produce apoptosis on H460 cell, which is consistent with the lack of MCL1 and XIAP downregulation in these cells.

-       - In Figure 2d, at difference of MTT the efficacy on colony formation seems to be better in SW900. Authors should comment this difference.

Indeed, in the MTT assays the A549 cells seem to be the most sensitive, but in the colony formation assays the concentration of apMNKQ2 is different in the three cell lines. Moreover, we have calculated the colony forming efficiency as 50.4%, 53% and 44% for A549, SW900 and H460 cells, respectively. Therefore, this difference in colony-forming capacity could explain the results obtained.  

-        - Similarly, in Figure 3, the aptamer alters cell migration only in A549 and SW900 and impairs MMPs only in A549. How the authors explain the difference between the three cells?

      apMNKQ2 decreases metalloproteases activity in A549 and SW900 cells but has no effect on H460 cells. As we mentioned in the discussion section, H460 cells harbor a PI3KCA mutation. The PI3K/AKT/mTOR pathway plays an important role in lung cancer cell invasion, specifically in the regulation of MMP-2 and MMP-9 (reference 55 in the new revised paper). These results suggest that the increased activity in the PI3K pathway in H460 cells could compensate for the effects of apMNKQ2, promoting the lack of an effect on MMP activity. We have included this explanation in discussion section (lines 557-561).

-    - In addition to cell migration assay, it would be useful to test aptamer ability to suppress cell invasion (can be done with Matrigel-coated transwell chambers).

Effectively, we carried out matrigel-coated transwell assays to check invasion, but we could not detect invasive cells. We tried to do it at two different concentrations of matrigel. At first, we used matrigel at stock concentration (9.8 mg/ml). Since we could not see any invasive cells in any of the cell lines, we used diluted matrigel at 2.5 mg/ml as has been reported in the literature for NSCLC cells. Nevertheless, at lower concentrations, we did not detect invasive cells. Accordingly, we checked the effect of apMNKQ2 on cell invasion through zymogram since this technique makes it possible to reveal the activity of metalloproteinases, which are involved in the extracellular matrix degradation during cell invasion processes.

-       - Why E-cadherin N-cadherin were only detected in A549 and SW900 cells? These markers should be expressed also by H460. I suggest the authors to add additional analyses (i.e RTqPCR) and/or include more EMT markers.

      The signal intensity of the bands corresponding to E-cadherin and N-cadherin in H460 cells is lower than in A549 and SW900. For this reason, we need a very high exposure to obtain bands in H460 cells, which does not allow us to obtain valid results for band quantification.

Regarding qPCR analyses, we consider the change in protein level to be more important than data on RNA since even if there are changes in RNA levels, they may not be reflected at the protein level.  

-       - The authors “do not rule out the possibility that apMNKQ2 may inhibit proliferation of lung cancer cells via cell cycle arrest”, this can be easily checked by FACS analyses. I suggest including this analyses

      We have carried out FACS analyses where we have observed a G1 and G2 phase arrest. We have included these data in results (new Figure 3a and lines 380-385) and discussion sections (lines 541-546).

Minor:

-       -    Figure 4a, 5a and b lack of statistical analyses. Please add significance

      We have added the p-value (p=0.08) in the new figure 5a (the previous figure 4a). There is no significance in the new figure 6 (the previous figure 5).

-     I suggest the authors check for few text errors. 

For example, Page 6 line 291: “In our previous work [37], four sequences were designed (apMNKQ1, apMNKQ2, 291 apMNKQ3 and apMNKQ4) from the aptamer apMNK2F against MNK1b and tested in breast cancer cell lines”

We have checked the text and corrected the errors.

-     It is unclear to me why some text part are in yellow, is it a revised manuscript?

      Yes, the highlighted text were the modifications from a first revision of the manuscript.

Round 2

Reviewer 2 Report (New Reviewer)

The authors addressed many of my concerns. I have only an additional concern on Figure 6. The authors claim that the aptamer “led to a reduction in both tumor volume (Figure 6a) and weight (Figure 6b)” but since they found that these differences are not significant, this cannot be stated. Authors need to change the text and discuss the necessity to better test the aptamer in vivo efficacy in the future.

Author Response

As requested, we have modified the text concern on Figure 6 in both results and discussion section. We have included and changed the following sentences, marked up using the “Track Changes”: In the results section, lines 487-488 “However, these results were not statistically significant, probably due to an unexpected slow tumor growth and/or the necessity of higher dose of aptamer.”

In the discussion section, lines 593-596 “We have shown that there is a downward trend in the tumor growth after treatment with apMNKQ2. Although tumor growth with xenografted A549 cells was slower than expected, we have obtained promising results that can be improved by increasing the dose of the aptamer”.

This manuscript is a resubmission of an earlier submission. The following is a list of the peer review reports and author responses from that submission.

Round 1

Reviewer 1 Report

The manuscripts explores development of a new strategy to treat lung cancer based on utilization of a therapeutic aptamer to inhibit MNK1. The group has previous experience whit aptamer development and also studies with MNKs. The subject is relevant, paper is interesting and need some clarifications, pointed bellow.  

1.      How does MNK1 is inactivated by the aptamer? Does it compete binding to a target or binds to the protein and inactivates protein?

2.      Material and metods does not describes Apt  sequence, and synthesis. This is an important information. Is it a DNA  ou RNA aptamer? Does it has modified bases? (o methyl, or 2’ fluorinated bases?) Please briefly describe in material and methods.

3.      In the figures 2,3,4  is shown a western blot pic and 2 densitometry graphs. Please clarify captions indicating “densitometry”, and also explaining samples  as p29 (is a control? Which control?) and Q2 (is the therapeutic aptamer?) Please check it in all figures.

4.      Figure 2a shows a western blot. Is it C a mock control? Why C was not quantified in figure 2b and 2c?

5.      How does aptamers are administrated in vivo? How does it target the right cell/tissue? Please comment in the manuscript.

6.      Please comment about systemic off-target toxicity of MNK1 inhibition using the aptamer-based-strategy.

7.      Figure5: I couldn’t find invivo experiments with SHAM animals and control aptamers. These controls are essential to understand invivo experimets with aptamers, since we may have non-specific effects due aptamers.

8.      In discussion authors mentioned aptamers lack toxicity. Which kind of assays were done to evaluate aptamer toxicity in comparison to other strategies?

Reviewer 2 Report

Submission ID: pharmaceutics-2144837

Type: Article

Title: An aptamer against MNK1 for non-small cell lung cancer treatment

Authors: Rebeca Carrión-Marchante, Celia Pinto-Díez, José Ignacio Klett-Mingo, Esther Palacios, Miriam Barragán-Usero1, María Isabel Pérez-Morgado, Manuel Pascual-Mellado, Laura Ruiz-Cañas, Bruno Sainz, Jr., Víctor M. González, M. Elena Martín

This manuscript describes the effects of an aptamer that binds specifically to MNK1 on lung cancer cell survival, proliferation, and invasion and on lung cancer cell growth and metastasis in vivo. However, it is controversial whether the effects of apMNKQ2, which was developed as an aptamer against MNK1, on the cells were caused by its effect on MNK1.

Previous and current reports of the authors have shown that apMNK2F or apMNKQ2, binds to MNK1, does not inhibit MNK1 kinase activity, can induce apoptosis in 80% of cancer cells by exposure to the aptamers for two days, and does not show toxicity when injected into the animals.

However, there are no reports on the effects of MNKs inhibitors or the knockdown of MNKs on cancer cells to induce apoptosis as strongly as apMNKQ2. Other groups have reported that inhibition of MNK1 kinase activity inhibits cancer cell proliferation and invasion, cancer cell xenografts, and metastasis in animals. There are many discrepancies between the results of these reports and the effects of aptamers on MNK1.

In addition, I think that the authors have not been able to prove whether the effect of the aptamers, including apMNK2F and apMNKQ2, on cancer cells is exerted by its inhibitory effect on MNK1 in previous reports and the present data.

While authors previously showed that apMNK2F is a potent inhibitor of the rabbit reticulocyte translation system. It was probably independent of MNK1.

Meanwhile, the authors previously showed that apMNK2F is a potent inhibitor of the rabbit reticulocyte translation system (Ref. 36). It is not thought to be an effect of apMNK2F specifically inhibiting MNK, and it would be toxic to any cells.

Considering these results, I have to conclude that the effects of apMNKQ2 on cancer cells shown in the authors' study are due to the off-target effects rather than the effects of apMNKQ2 on MNK1. Unless these questions are addressed, the premise of this article is lost.

In accepting the results of the present study, I believe it is necessary to show the results of the following considerations.

Major points

1. The authors need to show that knockdown of MNK1 by siRNAs in cancer cells (MDA-MB-321, MCF7, A549, AW900, SW460) in which apoptosis is induced by apMNKQ2 and apMNK2F, induces apoptosis to the same level as when apMNKQ2 is applied.

2. In addition, it would be even better if authors could develop a deletion mutant of MNK1 that acts as a dominant negative mutant to MNK1 and show that expression of the mutant induces apoptosis similar to the effect of apMNKQ2 in the above-mentioned cancer cells.

3. The authors need to show that apoptosis is rarely induced by apMNKQ2 in several normal cells, for example, embryonic fibroblast or primary cultured cells of tissues.

If the results of the above considerations are consistent with the results of the experiments using apMNKQ2, the premise of this paper will be proven to be correct, in that case, the following comments will be addressed.

The current study examines the effect of apMNKQ2 on lung cancer cells, and the results are almost identical to the previous report using breast cancer cells (Ref. 37). There are few new findings. However, if the authors address the above points and obtain data to resolve the discrepancies, it would be considered a sufficiently novel finding.

Minor point

Fig.1

The results of the MTT assay should also be shown after 24 hours. Because the authors measured the expression level of EMT markers and cell-cell adhesion molecules 24 h after apMNKQ2 was applied, the viability of cells should be indicated.

Fig. 2

The authors should show the total amount of PARP, or non-cleaved PARP, using other antibodies.

More discussion should describe why the effect of apMNKQ2 on the xenografts of cancer cells in vivo is so weak compared to its effect on cultured cancer cells.